# The impact of low- versus high-intensity surveillance cystoscopy on surgical care and cancer outcomes in patients with high-risk non-muscle-invasive bladder cancer (NMIBC)

**Michael E. Rezaee**[1,2], **Kristine E. Lynch**[3], **Zhongze Li**[4], **Todd A. MacKenzie**[4,5], **John D. Seigne**[1,6], **Douglas J. Robertson**[1,5], **Brenda Sirovich**[1,5], **Philip P. Goodney**[1,5], **Florian R. Schroeck**[1,2,5,6]*

1 White River Junction VA Medical Center, White River Junction, VT, United States of America, 2 Section of Urology Dartmouth Hitchcock Medical Center, Lebanon, NH, United States of America, 3 VA Salt Lake City Health Care System and University of Utah, Salt Lake City, UT, United States of America, 4 Biomedical Data Science Department, Geisel School of Medicine at Dartmouth College, Lebanon, NH, United States of America, 5 The Dartmouth Institute for Health Policy and Clinical Practice, Geisel School of Medicine at Dartmouth College, Lebanon, NH, United States of America, 6 Norris Cotton Cancer Center, Dartmouth Hitchcock Medical Center, Lebanon, NH, United States of America

* Florian.R.Schroeck@dartmouth.edu

**Data Availability Statement:** Data cannot be shared publicly because they contain potentially

## Abstract

### Purpose

To assess the association of low- vs. guideline-recommended high-intensity cystoscopic surveillance with outcomes among patients with high-risk non-muscle invasive bladder cancer (NMIBC).

### Materials & methods

A retrospective cohort study of Veterans Affairs patients diagnosed with high-risk NMIBC between 2005 and 2011 with follow-up through 2014. Patients were categorized by number of surveillance cystoscopies over two years following diagnosis: low- (1–5) vs. high-intensity (6 or more) surveillance. Propensity score adjusted regression models were used to assess the association of low-intensity cystoscopic surveillance with frequency of transurethral resections, and risk of progression to invasive disease and bladder cancer death.

### Results

Among 1,542 patients, 520 (33.7%) underwent low-intensity cystoscopic surveillance. Patients undergoing low-intensity surveillance had fewer transurethral resections (37 vs. 99 per 100 person-years; p<0.001). Risk of death from bladder cancer did not differ significantly by low (cumulative incidence [CIn] 8.4% [95% CI 6.5–10.9) at 5 years) vs. high-intensity surveillance (CIn 9.1% [95% CI 7.4–11.2] at 5 years, p = 0.61). Low vs. high-intensity surveillance was not associated with increased risk of bladder cancer death among patients with Ta (CIn 5.7% vs. 8.2% at 5 years p = 0.24) or T1 disease at diagnosis (CIn 10.2% vs. 9.1%

identifying and sensitive patient information. Data are available via the Veteran's IRB of Northern New England (contact via email: vhawrjresearchtask@va.gov) for researchers who meet the criteria for access to confidential Department of Veterans Affairs data. Data in the Department of Veterans Affairs Corporate Data Warehouse are collected for clinical purposes as part of the patient medical record. They contain potentially identifying and sensitive patient information and, therefore, cannot be shared. They can be accessed by any VA researcher through the Institutional Review Board process. Interested researchers can direct data access requests to the director of the Veteran's IRB of Northern New England, 215 N Main Street, White River Junction, VT 05009, phone 802-295-9363, email: vhawrjresearchtask@va.gov

**Funding:** FRS is supported by a Conquer Cancer Foundation Career Development Award and by the Dow-Crichlow Award of the Department of Surgery at the Dartmouth-Hitchcock Medical Center. PPG is supported by the Department of Veterans Affairs Health Services Research & Development (IIR 15-085, 1I01HX001880-01A2). The funding organizations had no role in the design and conduct of the study; collection, management, analysis, and interpretation of the data; preparation, review, or approval of the manuscript; and decision to submit the manuscript for publication.

**Competing interests:** The authors have declared that no competing interests exist.

at 5 years, p = 0.58). Among patients with Ta disease, low-intensity surveillance was associated with decreased risk of progression to invasive disease (T1 or T2) or bladder cancer death (CIn 19.3% vs. 31.3% at 5 years, p = 0.002).

## Conclusions

Patients with high-risk NMIBC undergoing low- vs. high-intensity cystoscopic surveillance underwent fewer transurethral resections, but did not experience an increased risk of progression or bladder cancer death. These findings provide a strong rationale for a clinical trial to determine whether low-intensity surveillance is comparable to high-intensity surveillance for cancer control in high-risk NMIBC.

## 1. Introduction

Bladder cancer is a common genitourinary malignancy with over 700,000 individuals diagnosed with the disease in the United States.[1] Non-muscle invasive bladder cancer (NMIBC) accounts for approximately 70–80% of bladder cancer diagnoses and is associated with substantially lower morbidity and mortality compared to muscle invasive disease.[2] After tissue diagnosis and staging, NMIBC can be risk stratified based on the probability of tumor recurrence or progression to muscle-invasive disease. Patients with high-risk lesions (high grade Ta, T1, or carcinoma in situ [CIS]) have up to an 80% risk of recurrence and up to a 50% risk of progression over 5 years.[2] As such, patients with high-risk NMIBC represent a unique population that may benefit from more aggressive treatment and surveillance practices.

After transurethral resection and possible intra-vesical therapy, patients with NMIBC undergo surveillance cystoscopy to assess for disease recurrence over time. Frequency of cystoscopy can vary as there are different surveillance recommendations for NMIBC amongst oncologic organizations.[3] For high-risk disease, the American Urological Association (AUA) recommends cystoscopy every 3–4 months for 2 years, every 6 months for an additional 2 years, and then annually.[4] The National Comprehensive Cancer Network recommends cystoscopy every 3–6 months for the first 2 years and then at "increasing intervals" afterwards.[5] However, these guidelines are based on minimal scientific data and mainly on expert opinion. Such limitations create gaps in evidence-based care and may contribute to patients undergoing more cystoscopies than needed.

We previously reported that approximately one-third of patients with high-risk NMIBC or CIS underwent fewer surveillance cystoscopies than recommended (i.e. low-intensity surveillance).[6] However, whether low-intensity surveillance among patients with high-risk NMIBC is associated with unfavorable outcomes is unknown. Our objective was to assess the association of low- vs. guideline-recommended high-intensity cystoscopic surveillance with key clinical outcomes, including frequency of transurethral resections, progression to invasive disease, and bladder cancer death.

## 2. Materials & methods

### 2.1 Study population

We conducted a retrospective cohort study of all Department of Veterans Affairs (VA) patients older than 65 years diagnosed with high-risk NMIBC according to the European Association of Urology guidelines (i.e., high grade Ta, CIS, or T1)[7] between 2005 and 2011 with follow-

up through 2014. As previously described,[6] a validated algorithm was used to identify 2,152 patients with newly diagnosed high-risk (high grade Ta, T1, or CIS) urothelial cell carcinoma of the bladder using national VA data. Pathologic information was extracted via validated natural language processing (NLP) algorithms.[8] Patients were excluded if they had missing covariates (n = 173). We also *a priori* excluded patients who died or had their last contact with the VA health system during the first two years after diagnosis, as this was the time period during which we measured intensity of surveillance (n = 437, 153 died from bladder cancer, 262 from other causes, 2 from unknown cause, and 20 had last contact within 2 years). The final analysis sample consisted of 1,542 patients.

## 2.2 Defining low- versus high-intensity cystoscopic surveillance

We defined intensity of cystoscopic surveillance according to the number of procedures received over a 2-year surveillance window. The surveillance window began with the NMIBC diagnosis date and ended 2 years after diagnosis or at the time of cystectomy, radiotherapy, or cancer recurrence, whichever came first. We did not enumerate cystoscopies after a cancer recurrence, because a recurrence increases risk for further recurrences, warrants additional resections, and thus "restarts the surveillance clock." Recurrences were identified from full-text pathology reports using the validated NLP algorithms.[8] We identified cystoscopy procedures using Current Procedural Terminology (CPT) codes as previously described[6] and categorized patients into those that received low- versus high-intensity surveillance based on current consensus guideline recommendations and the length of the surveillance window (Fig 1).[9,10,11] The surveillance window was evaluated by the following intervals: up to 5.5 months, 5.5 up to 9.5, 9.5 up to 13.5, 13.5 up to 17.5, 17.5 up to 21.5, and over 21.5 months. The rationale for these intervals was that surveillance cystoscopy at 4, 8, 12, 16, 20, and 24 months is recommended for high-risk bladder cancer by the AUA.[12] A 1.5 month grace period was allotted to allow for surveillance cystoscopies that were performed slightly later than recommended.

## 2.3 Outcomes

The outcomes of interest were number of transurethral resections, number of resections with and without cancer in the specimen, progression to invasive disease (T1 or T2), and bladder cancer death. First, we enumerated transurethral resections (including cystoscopy with biopsy) during a 2- to 9-year follow-up period using CPT and International Classification of Diseases (ICD), Ninth Revision, procedure codes as previously defined.[13] The follow-up period for our outcomes analysis started with the date of diagnosis and ended with the date of cystectomy, radiotherapy, cancer recurrence, death, last contact with the VA system, or at the end of the study (December 31, 2014), whichever came first. Resections performed less than 30 days apart were not considered in the enumeration of transurethral resections. This was done to avoid over-counting of re-resections or resections that were erroneously entered more than once in the medical record.

Second, the number of resections with and without cancer in the specimen was determined using validated NLP algorithms of VA pathology data.[8] Third, date of death was obtained from the VA Corporate Data Warehouse Vital status File, while cause of death was acquired from the National Death Index (NDI).[14] Bladder cancer death was defined using ICD, tenth revision, codes from the NDI.[13] Lastly, NLP algorithms were used to identify patients with Ta disease who progressed to invasive disease. Of note, progression to invasive disease included progression to any invasive disease, including invasion into the lamina propria (T1) or into the muscularis propria (T2). This was done because the NLP algorithms are limited in

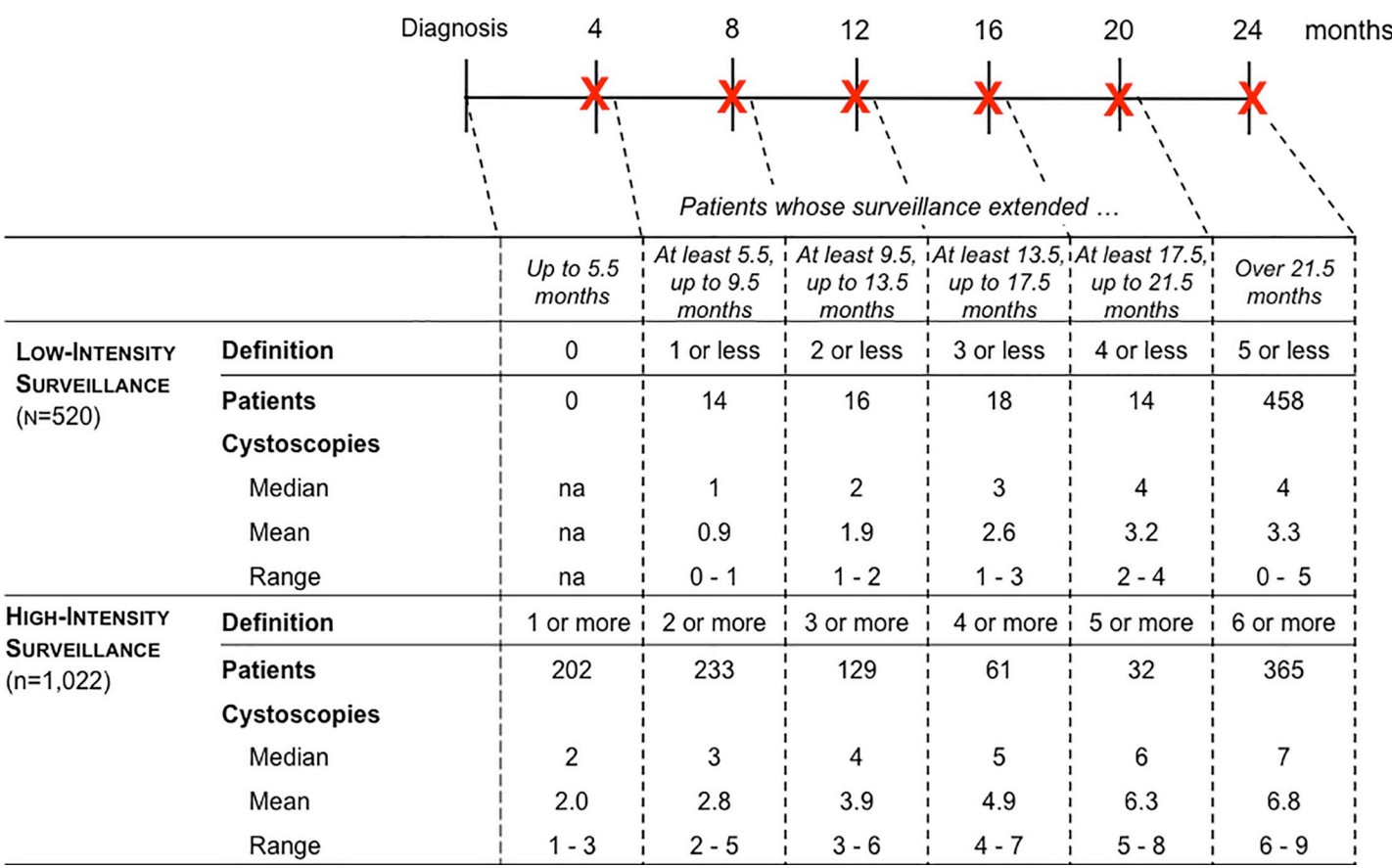

**Fig 1. Categorizing patients into low versus high-intensity surveillance based on consensus guideline recommendations and length of the surveillance window.**
[3] At the top of the figure, the timeline of the surveillance window is depicted in months. X denotes the recommended time of cystoscopy. A 1.5 month grace period was allotted to allow for surveillance cystoscopies that were done slightly later than recommended. For example, a patient followed for 9.5 months (second column) who underwent 0 or 1 cystoscopies was categorized as low-intensity surveillance, whereas a patient followed for 9.5 months who underwent 2 or more cystoscopies was categorized as high-intensity surveillance. In the table, the number of patients categorized into low versus high-intensity surveillance (overall and stratified by length of surveillance window), and number of surveillance cystoscopies is depicted.

their ability to differentiate between invasion into the lamina propria versus muscularis propria, likely due to the large variation in language used by pathologists to describe these findings.[8]

## 2.4 Statistical analysis

We first compared patient characteristics by low and high-intensity surveillance cystoscopy status using descriptive statistics. We then performed a series of propensity score adjusted analyses to assess the association between low- versus high-intensity surveillance and primary study outcomes. Propensity scores were calculated for each patient as the probability of undergoing low-intensity surveillance conditional on patient baseline characteristics listed in Table 1.[15,16]

First, propensity score adjusted Poisson regression was used to assess the association of low- versus high-intensity cystoscopic surveillance with frequency of transurethral resections, overall and with and without cancer in the specimen. Second, propensity adjusted Fine-Gray competing risks regression was used to assess the association of surveillance intensity with risk of bladder cancer death. These Fine-Gray regressions were also adjusted for receipt of

**Table 1. Baseline patient characteristics of 1,542 patients diagnosed with high-risk NMIBC stratified by low versus-high intesity cystoscopic surveillance.**

| | Total / All patients (n = 1542) | High intensity surveillance (n = 1022) | Low intensity surveillance (n = 520) | P-value [*] |
|---|---|---|---|---|
| Age (median, IQR) | 77 (66–95) | 76 (66–95) | 77 (66–94) | 0.04 |
| Age ≥80 (N, %) | 555 (36) | 343 (33.6) | 212 (40.8) | <0.01 |
| Male Sex (N, %)[**] | >1531 (>99.2) | >1011 (>98.9) | >509 (>97.8) | 0.33 |
| Race (N, %) | | | | |
| White[**] | >1273 (>82.5) | >856 (>83.7) | 414 (79.6) | 0.04 |
| Black | 113 (7.3) | 60 (5.9) | 53 (10.2) | |
| Asian[**] | 14 (0.9) | 11 (1.1) | <11 (<2.2) | |
| Hispanic[**] | 23 (1.5) | 16 (1.6) | <11 (<2.2) | |
| Native American[**] | <11 (<0.8) | <11 (<1.1) | <11 (<2.2) | |
| Unknown | 108 (7) | 68 (6.7) | 40 (7.7) | |
| Comorbidity (N, %) | | | | |
| 0 | 226 (14.7) | 142 (13.9) | 84 (16.2) | 0.39 |
| 1 | 403 (26.1) | 273 (26.7) | 130 (25) | |
| 2 | 404 (26.2) | 277 (27.1) | 127 (24.4) | |
| ≥3 | 509 (33) | 330 (32.3) | 179 (34.4) | |
| Nosos-p score [***] (median, IQR) | 1.6 (0.4–7.5) | 1.7 (0.5–7.5) | 1.5 (0.4–7.3) | 0.01 |
| Year of diagnosis (N, %) | | | | |
| 2005 | 33 (2.1) | 20 (2) | 13 (2.5) | 0.33 |
| 2006 | 182 (11.8) | 121 (11.8) | 61 (11.7) | |
| 2007 | 222 (14.4) | 140 (13.7) | 82 (15.8) | |
| 2008 | 267 (17.3) | 164 (16) | 103 (19.8) | |
| 2009 | 269 (17.4) | 184 (18) | 85 (16.3) | |
| 2010 | 312 (20.2) | 215 (21) | 97 (18.7) | |
| 2011 | 257 (16.7) | 178 (17.4) | 79 (15.2) | |
| Proportion living in ZIP code with ≥25% college graduates (N, %) | 645 (41.8) | 436 (42.7) | 209 (40.2) | 0.35 |
| Living in urban vs. rural area (N, %) | | | | |
| Urban | 929 (60.2) | 608 (59.5) | 321 (61.7) | 0.40 |
| Rural | 613 (39.8) | 414 (40.5) | 199 (38.3) | |
| Stage | | | | |
| Ta (high grade or associated with carcinoma in situ) | 599 (38.8) | 404 (39.5) | 195 (37.5) | 0.69 |
| T1 | 872 (56.5) | 570 (55.8) | 302 (58.1) | |
| Carcinoma in situ only | 71 (4.6) | 48 (4.7) | 23 (4.4) | |
| Carcinoma in situ | 330 (21.4) | 227 (22.2) | 103 (19.8) | 0.28 |
| Bladder Cancer Grade | | | | |
| Low[****] | 196 (12.7) | 123 (12) | 73 (14) | 0.26 |
| High | 1346 (87.3) | 899 (88) | 447 (86) | |
| Intravesical Therapy (N,%)[*****] | 859 (56) | 597 (58) | 262 (50) | <0.01 |

[*] From Chi-square test for categorical variable and Wilcoxon test for continuous variables whose median and IQR were presented. Missing observations were excluded for analysis.

[**] Exact numbers not shown to protect confidentiality.

[***] The Nosos-p score is a risk-adjustment score based on diagnosis codes, biographic information (including gender, date of birth, insurance coverage, race, marital status, VA priority (priority 1–9), and inclusion in a VA registry), drug prescription data and utilization costs. The "-p" indicates it is a prospective score, using data from one fiscal year to predict future health care utilization in the next fiscal year.

[****] Low-grade tumors were only included if they were T1 or associated with carcinoma in situ.

[*****] Not included in initial propensity score adjustment. However, all Fine-Gray models were adjusted for receipt on intravesical chemotherapy.

intravesical therapy during the surveillance period, which was ascertained using VA administrative and pharmacy data. Death from causes other than bladder cancer was modeled as a competing risk. This analysis step was stratified by Ta (high-grade or associated with CIS) *versus* T1 disease at the time of diagnosis, as well as surveillance intensity.

Third, a similar Fine-Gray regression model was used to assess the relationship between surveillance intensity and a combined outcome of progression to invasive disease (T1 or T2) or bladder cancer death among a subset of patients diagnosed with Ta disease. These outcomes were combined, because both of them represent an undesirable cancer outcome. As described above, progression to invasive disease (T1 or T2) could only be accurately measured among patients diagnosed with Ta disease due to NLP limitations. Thus, this model was run only for this subset of patients.

Finally, we performed sensitivity analyses to assess whether the exclusion of patients who died or had their last contact with the VA health system during the first two years after diagnosis affected our results. We re-calculated the propensity score on the cohort without implementing this exclusion and then refitted the Fine-Gray regression models to assess the relationship between intensity of surveillance and outcomes.

To standardize reporting and demonstrate effect size, numbers of resections were reported per 100 person-years. A p-value < 0.05 was used for statistical significance. The study was approved by the Veteran's Institutional Review Board of Northern New England (#897920) and the University of Utah Institutional Review Board (#00079402). Study data were not fully anonymized as pathology was extracted from full text pathology reports which included patient identifiers. Data were accessed between January 2015 and February 2020 via the secure VA Informatics and Computing Infrastructure. Informed written consent was waived for the study. Analyses were performed using SAS Enterprise Guide 7.15 and Stata v15.1.

## 3. Results

### 3.1 Low- versus high-intensity cystoscopic surveillance

Of the 1,542 patients diagnosed with high-risk NMIBC, 520 (33.7%) underwent low-intensity surveillance over the two-year follow up period. Patients who underwent low-intensity surveillance were less likely to be white (79.6% vs. 84.4%, p<0.01) compared to high-intensity surveillance patients (Table 1), and a lower proportion of them were treated with intravesical therapy (50.4% vs. 58.4%, p<0.01). Fig 1 displays the surveillance window associated with each guideline recommended cystoscopy interval time point (delineated by red X). The surveillance window was 21.5 to 24 months for 458 (88.1%) patients who underwent low-intensity surveillance compared to 365 (35.7%) patients who underwent high-intensity surveillance (p<0.001, Fig 1). The surveillance window tended to be longer among patients who underwent low-intensity surveillance, because a lower proportion of them experienced a recurrence within the first 2 years (69 of 520 (13.3%) vs 655 of 1022 patients (64.1%); p<0.001). Median number of cystoscopies was 4 and 7 for patients who underwent low and high-intensity surveillance over at least 21.5 months, respectively (Fig 1).

### 3.2 Intensity of cystoscopic surveillance & outcomes

After propensity score adjustment, patients who underwent low-intensity surveillance had almost 3-times fewer transurethral resections (37 [95% Confidence Interval (CI) 34–41] vs. 99 [95% CI 93–101] per 100 person-years; p<0.001) compared to those who underwent high-intensity surveillance. Similarly, low-intensity surveillance patients underwent approximately 3-times fewer resections with cancer in the specimen (28 [95% CI 25–31] vs. 77 [95% CI 72–

83) per 100 person-years, p<0.001), and 2-times fewer resections without cancer in the specimen (7.5 [95% CI 6.5–8.7] vs. 16 [95% CI 15–18] per 100 person-years; p<0.001).

There were a total of 143 bladder cancer deaths during a median follow-up of 4.6 years. After adjustment for propensity score and receipt of intravesical therapy, risk of death from bladder cancer did not differ significantly by low (cumulative incidence [CIn] 8.4% [95% CI 6.5–10.9] at 5 years) vs. high-intensity surveillance (CIn 9.1% [95% CI 7.4–11.2] at 5 years, p = 0.61). Patients were then stratified by Ta *versus* T1 disease at the time of diagnosis, as well as surveillance intensity. Low vs. high-intensity surveillance was not associated with risk of bladder cancer death among patients with Ta (CIn 5.7% vs. 8.2% at 5 years p = 0.24) or with T1 disease at diagnosis (CIn 10.2% vs. 9.1% at 5 years, p = 0.58, Fig 3A). Among patients with Ta disease, low-intensity surveillance was associated with decreased risk of progression to invasive disease (T1 or T2) or bladder cancer death (CIn 19.3% vs. 31.3% at 5 years, p = 0.002, Fig 3B).

In sensitivity analyses, now including any patients who died or had their last contact with the VA health system during the first two years after diagnosis, we found very similar results. Again, risk of death from bladder cancer did not differ significantly by low (CIn 13.1% [95% CI 10.7–16.0] at 5 years) vs. high-intensity surveillance (CIn 14.9% [95% CI 13.1–16.9] at 5 years, p = 0.27). Results among patients stratified by Ta *versus* T1 disease were also essentially unchanged, with no apparent differences in bladder cancer death between low- and high intensity surveillance (Fig 2).

## 4. Discussion

We found that low-intensity surveillance was associated with 3-times fewer total transurethral resections in patients who underwent low versus guideline-recommended high-intensity cystoscopic surveillance of high-risk NMIBC (high grade Ta, T1, or CIS). There was no difference in risk of death from bladder cancer by surveillance intensity. Among a subset of patients with Ta disease, low-intensity surveillance was associated with decreased risk of progression to invasive disease (T1 or T2) or bladder cancer death.

To the best of our knowledge, we are the first to assess the association of low- vs. guideline-recommended high-intensity cystoscopic surveillance with outcomes in high-risk NMIBC patients. Our results suggest that there is no increased risk of disease progression or bladder cancer death in older patients with high-risk NMIBC who undergo low-intensity surveillance (Fig 3A). While the optimal surveillance intensity for high-risk NMIBC is unknown, low-intensity cystoscopic surveillance may be reasonable based on our results. It was associated with 3-times fewer transurethral resections compared to high-intensity surveillance (Fig 4). Thus, one of the benefits of low-intensity surveillance may be decreasing a patient's exposure to peri-operative risks and complications associated with repetitive cystoscopy and transurethral resection, including urinary tract infection, hematuria, and anesthesia complications. [17,18] In addition, low-intensity surveillance has the potential to decrease unnecessary testing and transurethral resections in patients with high-risk NMIBC given the difference in the number of transurethral resections we observed between the low- and high-intensity surveillance groups.

Among a subset of patients diagnosed with Ta disease, low-intensity surveillance was associated with a decreased risk of progression to invasive disease (T1 or T2) or bladder cancer death (Fig 3B). One would expect that low-intensity surveillance would delay the detection of recurrences and thus should increase the risk of progression to invasive disease (T1 or T2) or bladder cancer death. We found the opposite. Given the observational nature of our data, it is possible that urologists managed certain high-risk NMIBC patients with low-intensity

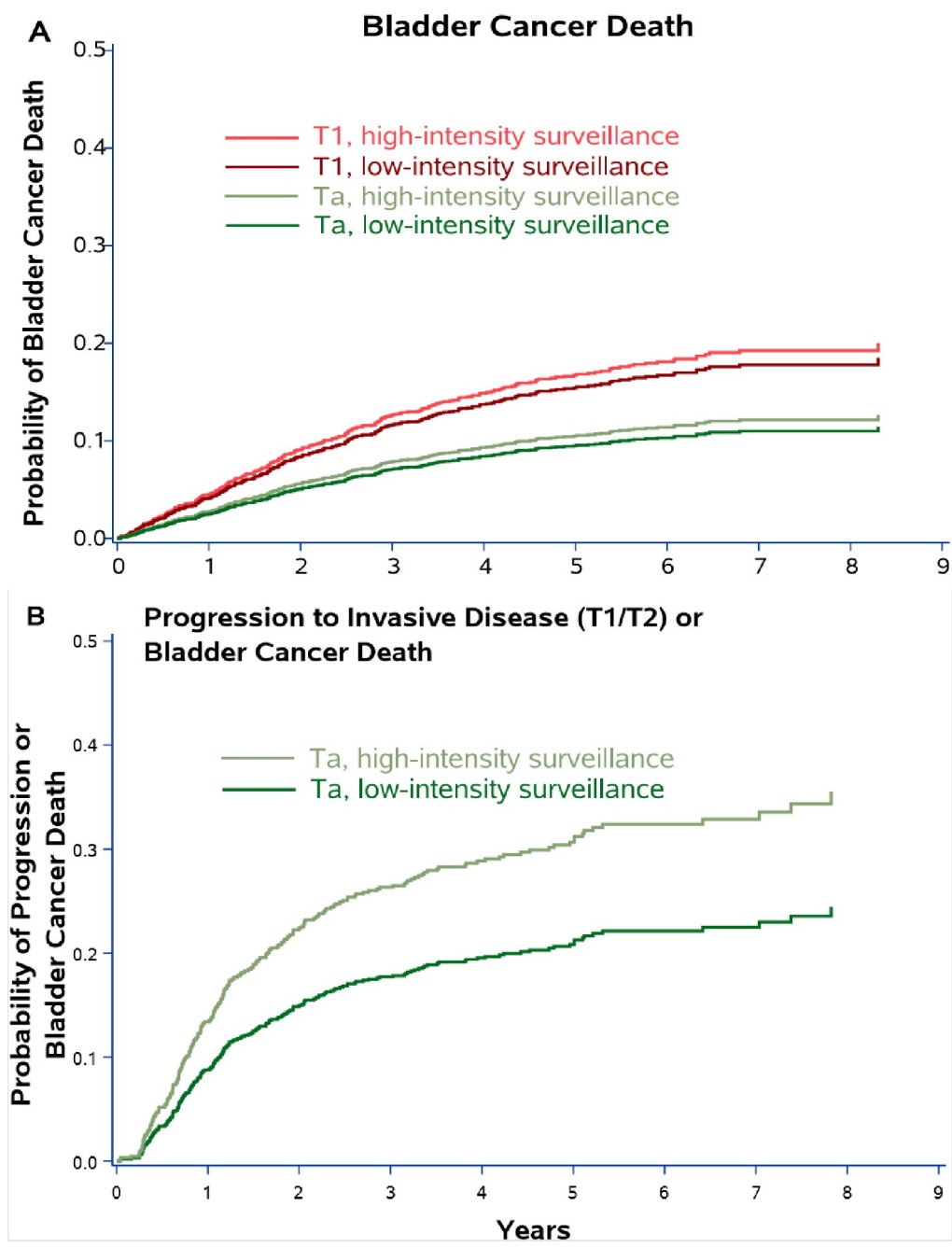

**Fig 2. Sensitivity analyses, now including any patients who died or had their last contact with the VA health system during the first two years after diagnosis.** Cumulative incidence plots showing the probability of 1) bladder cancer death by Ta *versus* T1 disease and by cystoscopic surveillance intensity (Panel A) and 2) progression to invasive disease (T1 or T2) or bladder cancer death among those with Ta disease (Panel B). Data are from Fine and Gray competing risk models adjusted for propensity score and receipt of intravesical therapy with death from other causes modeled as a competing risk.

surveillance based on intra-operative and patient characteristics not measured in our study. For example, tumor size and number of tumors were not available in our pathologic data as this was abstracted via NLP from full text bladder cancer pathology reports.[8] Urologists may have followed solitary or small high-risk lesions less intensely compared to multi-focal or

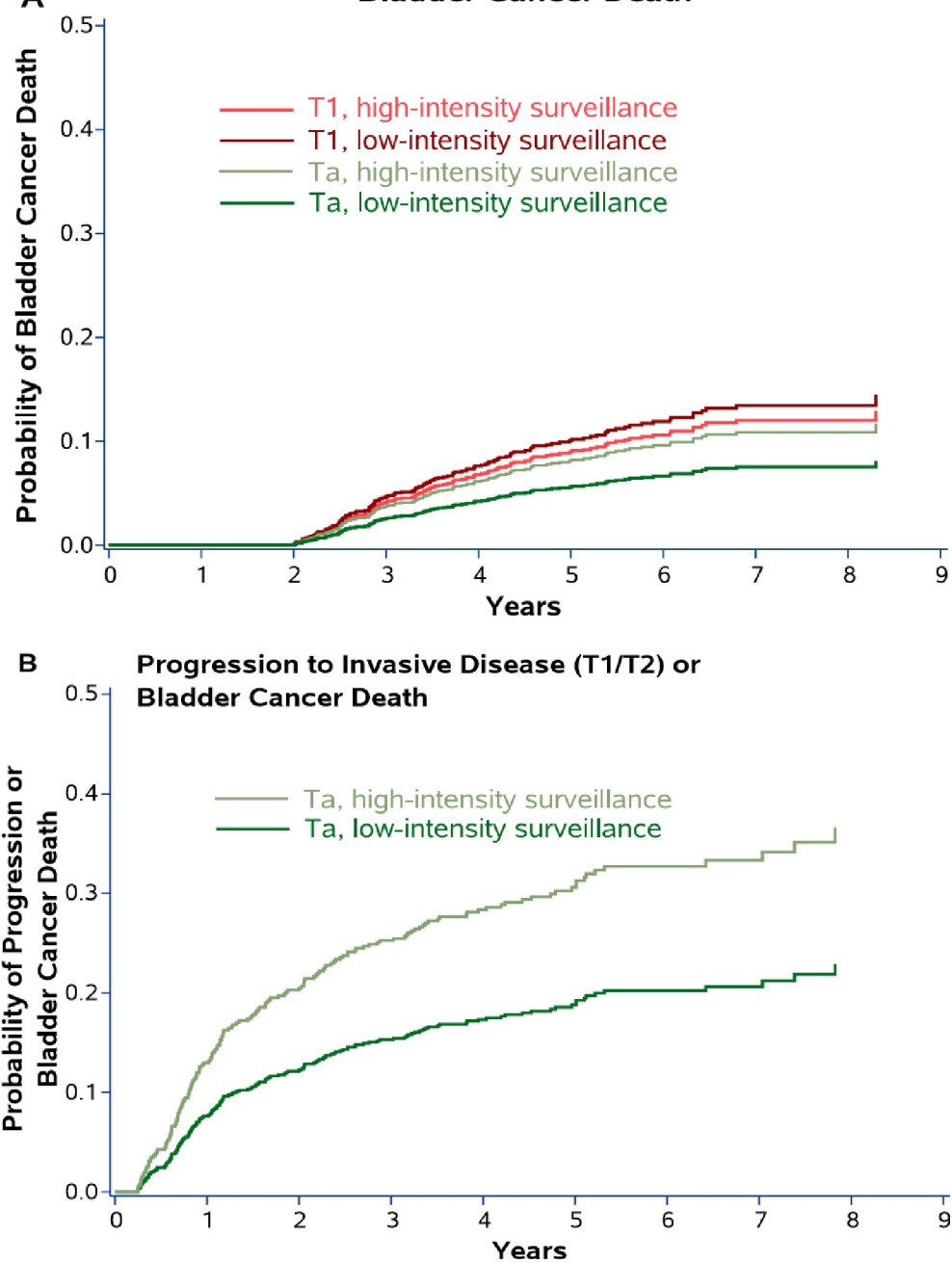

**Fig 3.** Cumulative incidence plots showing the probability of 1) bladder cancer death by Ta *versus* T1 disease and by cystoscopic surveillance intensity (Panel A) and 2) progression to invasive disease (T1 or T2) or bladder cancer death among those with Ta disease (Panel B). Data are from Fine and Gray competing risk models adjusted for propensity score and receipt of intravesical therapy with death from other causes modeled as a competing risk.

larger lesions, resulting in unmeasured additional risk-stratification by urologists. Thus, patients who underwent low-intensity surveillance may have had an inherently lower risk of progression than those undergoing high-intensity surveillance (*i.e.* unmeasured confounding). This may also suggest that distinct subpopulations of high-risk bladder cancer patients exist

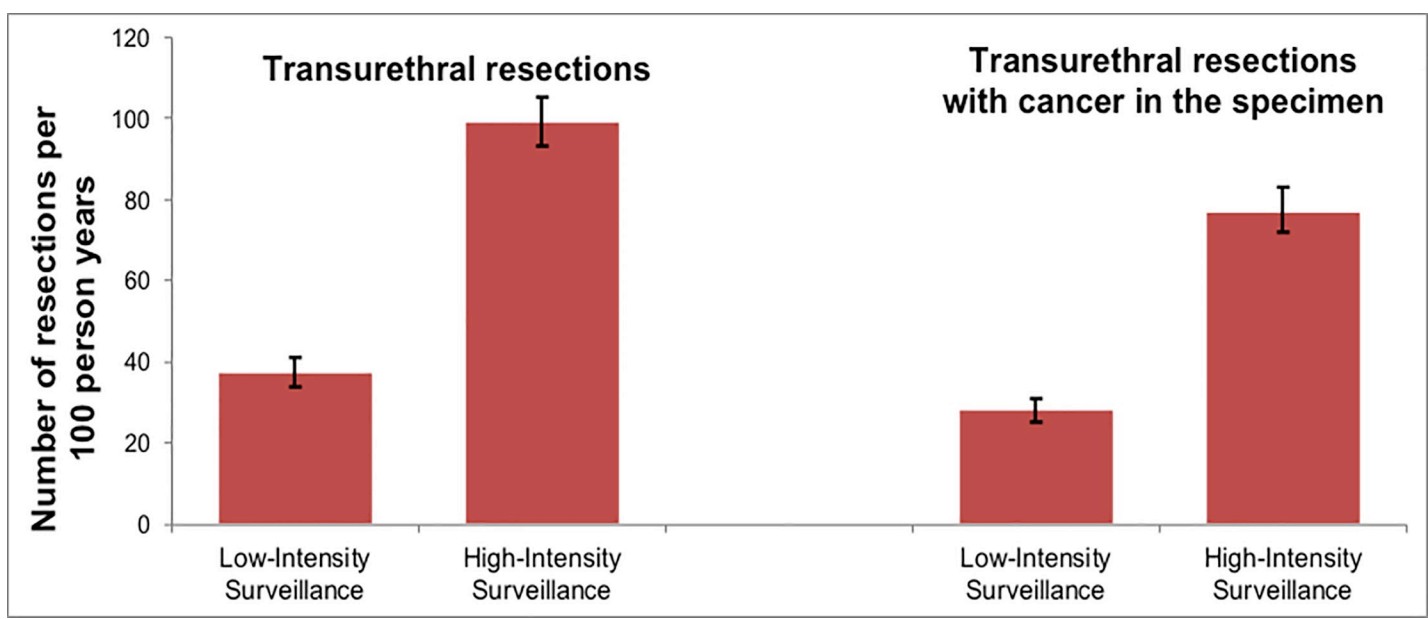

**Fig 4. Number of total transurethral resctions and resections with cancer in the pathology specimen by low versus high-intensity surveillance.** Patients who underwent low-intensity surveillance experienced 3-times fewer total transurethral resections and resection with cancer in the specimen compared to high-intensity surveillance patients.

that can be managed more similarly to patients with low-risk disease, which should be evaluated in future prospective studies. Further, high-intensity surveillance (looking more often) likely leads to earlier detection of progressive disease. In a time-to-event analysis, time to disease progression or death would then be substantially shorter in patients undergoing high-intensity surveillance, resulting in a higher calculated risk of progression. We believe that the higher risk of progression among patients undergoing high-intensity surveillance is likely due to a combination of unobserved confounding and earlier detection.

There are additional limitations of this study to consider. First, our study was performed in VA patients older than 65 years of age, which may limit the generalizability of our results to the general population. VA patients have more co-morbidities and poorer socioeconomic status compared to the general population.[19] However, the majority of new bladder cancer cases in the United States occur in men older than 65 years of age,[20] which our cohort adequately represents. Second, we had a moderate length of follow up (median 4.6 years) to assess for differences in outcomes. Studies examining longer-term follow up are required to determine if our findings persist over time. Third, as discussed above, it's difficult to understand why some patients underwent low-intensity surveillance not in accordance with AUA guidelines. In prior work, we identified African American race, no comorbidity, and male provider gender as factors associated with underuse of surveillance.[21] However, there are likely a number of other unmeasured confounders in our study that influence surveillance intensity. Lastly, we did not assess the potential impact of adjunct testing/imaging (e.g. cross-sectional imaging, ultrasound, biomarkers, or cytology) that may have influenced surveillance cystoscopy practices.

The primary strengths of our study include a large sample size and examination of outcomes that are important to both patients and physicians. Further, the use of full-text pathology reports provided a level of detail that allowed us to ascertain progression to invasive disease. This overcomes a weakness of commonly used oncological databases, such as the

National Cancer Database and Surveillance Epidemiology and End Results-Medicare database, which do not contain specific detail on resections performed for suspected cancer recurrence. [22,23]

Our study has important implications as the findings suggest that less intensive surveillance might be reasonable for patients with high-risk NMIBC, in contrary to what is currently recommended by major oncologic organizations.[3] Evidence supporting these recommendations is insufficient, which is explicitly stated in the AUA guideline.[4,12] Perhaps due to the lack of strong evidence, adherence to these surveillance recommendations is known to be poor.[24,25] In the present study, 33.7% high-risk NMIBC patients underwent less frequent cystoscopy than recommended, but did not have an increased risk of progression to bladder cancer death. These cumulative findings suggest that the optimal cystoscopic surveillance intensity for patients with high-risk NMIBC needs to be established using rigorous research methods.

## 5. Conclusion

Patients with high-risk NMIBC undergoing low-intensity cystoscopic surveillance underwent fewer transurethral resections than those with high-intensity surveillance, but did not experience an increased risk of disease progression or bladder cancer death. These findings suggest that less intensive surveillance might be reasonable for patients with high-risk NMIBC. However, given the retrospective nature of this study, our findings are subject to unmeasured confounding. Thus, we do not advocate for lower intensity surveillance of high-risk NMIBC based on our findings. Rather, we believe our study provides a strong rationale for a future randomized trial to assess whether low-intensity surveillance of patients with high-risk NMIBC is comparable to high-intensity surveillance in terms of cancer control.

## Acknowledgments

This study was supported using resources and facilities at the White River Junction Department of Veterans Affairs (VA) Medical Center, the VA Salt Lake City Health Care System, and the VA Informatics and Computing Infrastructure (VINCI), VA HSR RES 13–457. Support for VA and Centers for Medicare & Medicaid Services data is provided with support from the VA Information Resource Center, Project Numbers SDR 02–237 and 98–004. We acknowledge programming assistance by Benjamin Viernes, MPH.

**Disclaimer:** Opinions expressed in this manuscript are those of the authors and do not constitute official positions of the U.S. Federal Government or the Department of Veterans Affairs.

## Author Contributions

**Conceptualization:** Kristine E. Lynch, Todd A. MacKenzie, John D. Seigne, Douglas J. Robertson, Brenda Sirovich, Philip P. Goodney, Florian R. Schroeck.

**Data curation:** Michael E. Rezaee, Kristine E. Lynch, Zhongze Li, Todd A. MacKenzie, John D. Seigne, Douglas J. Robertson, Florian R. Schroeck.

**Formal analysis:** Michael E. Rezaee, Kristine E. Lynch, Zhongze Li, Florian R. Schroeck.

**Funding acquisition:** Florian R. Schroeck.

**Investigation:** Kristine E. Lynch, Todd A. MacKenzie, John D. Seigne, Douglas J. Robertson, Brenda Sirovich, Florian R. Schroeck.

**Methodology:** Michael E. Rezaee, Kristine E. Lynch, Zhongze Li, Todd A. MacKenzie, Douglas J. Robertson, Brenda Sirovich, Philip P. Goodney, Florian R. Schroeck.

**Project administration:** Brenda Sirovich, Florian R. Schroeck.

**Resources:** Florian R. Schroeck.

**Software:** Florian R. Schroeck.

**Supervision:** Florian R. Schroeck.

**Validation:** Philip P. Goodney, Florian R. Schroeck.

**Visualization:** Florian R. Schroeck.

**Writing – original draft:** Michael E. Rezaee, Florian R. Schroeck.

**Writing – review & editing:** Michael E. Rezaee, John D. Seigne, Florian R. Schroeck.

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
