## [Decision Letter · Decision Letter 0]

5 Feb 2020

PONE-D-20-01946

The impact of low- versus high-intensity surveillance cystoscopy on surgical care and cancer outcomes in patients with high-risk non-muscle-invasive bladder cancer (NMIBC)

PLOS ONE

Dear Dr. Rezaee,

Both reviewers concur that the topic is important and the issue raised is not settled. They also concur that such an important question will not be answered conclusively by a retrospective study. After careful consideration, we feel that it has merit but does not fully meet PLOS ONE’s publication criteria as it currently stands. Therefore, we invite you to submit a revised version of the manuscript that addresses the points raised during the review process. You should respond to all the comments of the referees providing the requested information and should tone down your conclusions by taking into consideration the limitations derived from the study design and the possible imbalances in the two study groups.

We would appreciate receiving your revised manuscript within 3 months. To enhance the reproducibility of your results, we recommend that if applicable you deposit your laboratory protocols in protocols.io, where a protocol can be assigned its own identifier (DOI) such that it can be cited independently in the future. For instructions see: http://journals.plos.org/plosone/s/submission-guidelines#loc-laboratory-protocols

We look forward to receiving your revised manuscript.

Kind regards,

Francisco X. Real

Academic Editor

PLOS ONE

Journal Requirements:

1. In the ethics statement in the manuscript and in the online submission form, please provide additional information about the patient records used in your retrospective study, including: a) whether all data were fully anonymized before you accessed them; b) the date range (month and year) during which patients' medical records were accessed; and c) the source of the medical records analyzed in this work (e.g. hospital, institution or medical center name). If patients provided informed written consent to have data from their medical records used in research, please include this information.

2. Thank you for including the following funding statement in your manuscript; " FRS is supported by a Conquer Cancer Foundation Career Development Award and by the Dow-Crichlow Award of the Department of Surgery at the DartmouthHitchcock Medical Center. PPG is supported by the Department of Veterans Affairs Health Services Research & Development (IIR 15-085, 1I01HX001880-01A2). The funding organizations had no role in the design and conduct of the study; collection, management, analysis, and interpretation of the data; preparation, review, or approval of the manuscript; and decision to submit the manuscript for publication."

We note that you have provided funding information that is not currently declared in your Funding Statement. However, funding information should not appear in your manuscript. We will only publish funding information present in the Funding Statement section of the online submission form.

"none"

3. Thank you for including your competing interests statement; "none"

Reviewers' comments:

Reviewer's Responses to Questions

**Comments to the Author**

1. Is the manuscript technically sound, and do the data support the conclusions?

Reviewer #1: Partly

Reviewer #2: Partly

2. Has the statistical analysis been performed appropriately and rigorously? 

Reviewer #1: Yes

Reviewer #2: Yes

3. Have the authors made all data underlying the findings in their manuscript fully available?

Reviewer #1: Yes

Reviewer #2: Yes

4. Is the manuscript presented in an intelligible fashion and written in standard English?

Reviewer #1: Yes

Reviewer #2: Yes

5. Review Comments to the Author

Reviewer #1: This is a retrospective study on a specific database of patients with high-risk non-muscle-infiltrating bladder cancer. This database also provides reliable follow-up (only 195 lost for follow-up of a total of more than 2100 patients). In this database, the diagnostic profitability and oncological results of establishing a narrow cystoscopic follow-up (one cystoscopy every 4 months minimum) or more lax (less than one cystoscopy every 4 months) are studied, finding that there are no differences between the two follow-ups.

It is an attractive study, with results that push to a more lax follow up, and tries to fill a real world clinical gap. However, the work has the following limitations:

- The exclusion of patients deceased by bladder cancer during the first two years of follow-up must be better explained. It is likely that these patients were part of a closer follow-up and more tumor resections were performed. I would propose to include those patients who died from bladder cancer who had undergone a review with at least one cystoscopy without tumor recurrence. In this way, an attempt is made to exclude patients in whom a non-infiltrating bladder tumor was mistakenly diagnosed in the beginning. To know in which group they are placed, the number of cystoscopies performed is divided into the time spent from initial resection to the diagnosis of the tumor (less than 3 cystoscopies/year or more than 3c/year)

- As the authors point out, the number of tumors is not reflected, which can be an important confounding factor in the cystoscopy rhythm and the results obtained.

- It is not specified whether the follow-up was replaced by other means (ultrasound, CAT Scan…) or if other urothelial tumors (upper urinary tract) were detected during the follow up. These results may influence the results presented.

- The percentage of patients over 80 years in the low intensity group is higher, and that may have influenced in not performing resection of asymptomatic tumors detected.

- Also, it has to be said that some low risk tumours may have been detected, but not operated or biopsied. This may influence the results presented.

- The grouping between non-infiltrating tumors (Ta) and CIS is not correct: High grade Ta may have around a 10% risk of progression, while the CIS can reach 50% (figure 3). This analysis should be reconsidered only with Ta vs T1 if the number of patients with CIS is too small.

Reviewer #2: The authors have assessed the association of low- vs. guideline-recommended high-intensity

cystoscopic surveillance with outcomes among patients with high-risk NMIBC analyzing a retrospective cohort study of Veterans Affairs patients (> 75) diagnosed with high-risk disease between 2005 and 2011 with follow-up through 2014

They found that patients with high-risk NMIBC undergoing low- vs. high-intensity cystoscopic surveillance underwent fewer transurethral resections, but did not experience an increased risk of progression or bladder cancer death

General comments

The evidence behind the frequency of cystoscopies in the follow-up of NMIBC is particularly poor and not supported by any level one evidence. Numerous studies including a previous one by the same group of authors (but mainly in low grade disease) have clearly outlined a major risk of overuse for various reasons.

Therefore the attempt to correlate the frequency of cystoscopic examinations by the authors should be commended and addresses a real unmet need.

Although the results are nicely presented and the authors have done their very best to honestly analyze and criticize their findings nevertheless the reader is left with the impression that the low and high frequency groups may not have been similar, therefore weakening any robust conclusions.

1.The confounder and possible bias is patients with low intensity might have had a better prognosis and less recurrences (even the high risk NMIBC group is heterogenous as outlined by the authors in their discussion); so the treating urologist might have had the tendency to space follow-up.

2. After propensity score adjustment, patients who underwent low-intensity surveillance had almost 3-times fewer transurethral resections compared to those who underwent high-intensity surveillance: In high grade NMIBC, especially given the fact that these patients usually receive BCG leading to bladder inflammation, it is reasonable to assume that more frequent cystos led to a higher chance that the urologists would see something she/he didn't like, decided to err on side of caution and therefore resect, isn't it?

3.The surveillance window tended to be longer among patients who underwent low-intensity surveillance,

because a lower proportion of them experienced a recurrence within the first 2 years (69

of 520 (13.3%) vs 655 of 1022 patients (64.1%); p<0.001).

This is honestly difficult to reconcile and grasp. It seems that despite the matching, groups were imbalanced in terms of risk of recurrence/progression.

4.Among patients with non-invasive disease (high grade Ta/CIS), low-intensity surveillance was associated with decreased risk of progression to invasive disease (T1 or T2) or bladder cancer death (CIn 19.3% vs. 31.4% at 5 years, p=0.002, Figure Panel 3B)- This is again very difficult to understand?? Why would any lower surveillance be associated with decreased risk of progression unless low intensity surveillance was performed in lower risk populations - the authors acknowledge that it is possible that urologists

managed certain high-risk NMIBC patients with low-intensity surveillance based on intra-operative and patient characteristics not measured in their study.Urologists may have followed solitary or small high-risk lesions less intensely compared to multi-focal or larger tumors

In conclusion, this is a nicely written paper on an important topic but the conclusions are not fully supported by the data presented and would benefit from being tempered.

6. PLOS authors have the option to publish the peer review history of their article (what does this mean?). If published, this will include your full peer review and any attached files.

Reviewer #1: No

Reviewer #2: No

---

## [Author Response · Author response to Decision Letter 0]

27 Feb 2020

Response to Reviewers

Journal Requirements:

1. In the ethics statement in the manuscript and in the online submission form, please provide additional information about the patient records used in your retrospective study, including: a) whether all data were fully anonymized before you accessed them; b) the date range (month and year) during which patients' medical records were accessed; and c) the source of the medical records analyzed in this work (e.g. hospital, institution or medical center name). If patients provided informed written consent to have data from their medical records used in research, please include this information.

We have added the following to the ethics statement in the manuscript: 

“Study data were not fully anonymized as pathology was extracted from full text pathology reports which included patient identifiers. Data were accessed between January 2015 and February 2020 via the secure VA Informatics and Computing Infrastructure. Informed written consent was waived for the study.”

2. Thank you for including the following funding statement in your manuscript; " FRS is supported by a Conquer Cancer Foundation Career Development Award and by the Dow-Crichlow Award of the Department of Surgery at the DartmouthHitchcock Medical Center. PPG is supported by the Department of Veterans Affairs Health Services Research & Development (IIR 15-085, 1I01HX001880-01A2). The funding organizations had no role in the design and conduct of the study; collection, management, analysis, and interpretation of the data; preparation, review, or approval of the manuscript; and decision to submit the manuscript for publication."

We note that you have provided funding information that is not currently declared in your Funding Statement. However, funding information should not appear in your manuscript. We will only publish funding information present in the Funding Statement section of the online submission form.

Please remove any funding-related text from the manuscript and let us know how you would like to update your Funding Statement. 

Thank you. We have removed the funding statement from the manuscript. We would like our funding statement to read as:

“FRS is supported by a Conquer Cancer Foundation Career Development Award and by the Dow-Crichlow Award of the Department of Surgery at the Dartmouth-Hitchcock Medical Center. PPG is supported by the Department of Veterans Affairs Health Services Research & Development (IIR 15-085, 1I01HX001880-01A2). The funding organizations had no role in the design and conduct of the study; collection, management, analysis, and interpretation of the data; preparation, review, or approval of the manuscript; and decision to submit the manuscript for publication.”

3. Thank you for including your competing interests statement; "none"

This manuscript was submitted by the first author: Michael E. Rezaee (ORCID iD 0000-0002-6236-8682). However, we would greatly appreciate it if you could list the senior author, Florian R. Schroeck (ORCID iD 0000-0002-1860-2611) as the final corresponding author. 

Reviewers' comments:

Reviewer's Responses to Questions

Comments to the Author

1. Is the manuscript technically sound, and do the data support the conclusions?

Reviewer #1: Partly

Reviewer #2: Partly

2. Has the statistical analysis been performed appropriately and rigorously?

Reviewer #1: Yes

Reviewer #2: Yes

3. Have the authors made all data underlying the findings in their manuscript fully available?

Reviewer #1: Yes

Reviewer #2: Yes

For clarification:

Data in the Department of Veterans Affairs Corporate Data Warehouse are collected for clinical purposes as part of the patient medical record. They contain potentially identifying and sensitive patient information and, therefore, cannot be shared. They can be accessed by any VA researcher through the Institutional Review Board process. Interested researchers can direct data access requests to the director of the Veteran's IRB of Northern New England, 215 N Main Street, White River Junction, VT 05009, phone 802-295-9363, email: vhawrjresearchtask@va.gov

Data cannot be shared publicly because they contain potentially identifying and sensitive patient information. Data are available via the Veteran's IRB of Northern New England (contact via email: vhawrjresearchtask@va.gov) for researchers who meet the criteria for access to confidential Department of Veterans Affairs data

4. Is the manuscript presented in an intelligible fashion and written in standard English?

Reviewer #1: Yes

Reviewer #2: Yes

5. Review Comments to the Author

Reviewer #1: This is a retrospective study on a specific database of patients with high-risk non-muscle-infiltrating bladder cancer. This database also provides reliable follow-up (only 195 lost for follow-up of a total of more than 2100 patients). In this database, the diagnostic profitability and oncological results of establishing a narrow cystoscopic follow-up (one cystoscopy every 4 months minimum) or more lax (less than one cystoscopy every 4 months) are studied, finding that there are no differences between the two follow-ups.

It is an attractive study, with results that push to a more lax follow up, and tries to fill a real world clinical gap. However, the work has the following limitations:

- The exclusion of patients deceased by bladder cancer during the first two years of follow-up must be better explained. It is likely that these patients were part of a closer follow-up and more tumor resections were performed. I would propose to include those patients who died from bladder cancer who had undergone a review with at least one cystoscopy without tumor recurrence. In this way, an attempt is made to exclude patients in whom a non-infiltrating bladder tumor was mistakenly diagnosed in the beginning. To know in which group they are placed, the number of cystoscopies performed is divided into the time spent from initial resection to the diagnosis of the tumor (less than 3 cystoscopies/year or more than 3c/year)

We decided a priori to exclude patients who died or had their last contact with the VA health system during the first two years after diagnosis, as this was the time period during which we measured intensity of surveillance. These patients had a limited opportunity to be exposed to cystoscopic surveillance, which was the primary exposure being evaluated in the study. However, we understand the reviewer’s concern regarding this decision. To address this, we have now performed additional sensitivity analyses: We re-ran all models and re-evaluated bladder cancer death as well as progression to invasive bladder cancer or bladder cancer death without excluding these patients. Also, all patients included in the study had at least one surveillance cystoscopy. 

We have added the following to the methods in the analyses section:

“Finally, we performed sensitivity analyses to assess whether the exclusion of patients who died or had their last contact with the VA health system during the first two years after diagnosis affected our results. We re-calculated the propensity score on the cohort without implementing this exclusion and then refitted the Fine-Gray regression models to assess the relationship between intensity of surveillance and outcomes.”

We found the following and added this to the Results section along with new Figure 4:

“In sensitivity analyses, now including any patients who died or had their last contact with the VA health system during the first two years after diagnosis, we found very similar results. Again, risk of death from bladder cancer did not differ significantly by low (CIn 13.1% [95% CI 10.7-16.0] at 5 years) vs. high-intensity surveillance (CIn 14.9% [95% CI 13.1-16.9] at 5 years, p=0.27). Results among patients stratified by Ta versus T1 disease were also essentially unchanged, with no apparent differences in bladder cancer death between low- and high intensity surveillance (Figure 4).”

- As the authors point out, the number of tumors is not reflected, which can be an important confounding factor in the cystoscopy rhythm and the results obtained.

We agree that this is an important limitation of the study that is discussed in the discussion section of the paper. 

- It is not specified whether the follow-up was replaced by other means (ultrasound, CAT Scan…) or if other urothelial tumors (upper urinary tract) were detected during the follow up. These results may influence the results presented.

Per AUA guidelines patients with high-risk NMIBC should undergo surveillance cystoscopy at specific time points. Upper tract evaluation with a CT Urogram or bilateral retrogrades can be considered every 1-2 years for high-risk disease. We did not evaluate the potential role of other modalities for disease surveillance (bladder ultrasound, CT Urogram, biomarkers, etc.) because direct visualization with cystoscopy is the standard of care for disease surveillance. It’s possible that other modalities could have assisted in identifying disease recurrence. However, it’s unlikely that a significant percentage of patients in our study underwent non-guideline recommended imaging/testing for surveillance purposes. We have added this as a potential limitation of the study to the discussion section: 

“Lastly, we did not assess the potential impact of adjunct testing/imaging (e.g. cross-sectional imaging, ultrasound, biomarkers, or cytology) that may have influenced surveillance cystoscopy practices. ”

- The percentage of patients over 80 years in the low intensity group is higher, and that may have influenced in not performing resection of asymptomatic tumors detected.

There are clearly observed differences between patients undergoing low- versus high-intensity cystoscopic surveillance. As can be seen in Table 1, this includes differences in age, among others. All of these differences were accounted for in our propensity score and propensity score adjusted models. 

- Also, it has to be said that some low risk tumours may have been detected, but not operated or biopsied. This may influence the results presented.

It would be atypical for a patient with a history of high-risk NMIBC to have a tumor recurrence seen on surveillance cystoscopy, but not have the tumor resected or biopsied because it visually appears low grade. Even if this happened in rare instances, low-grade tumors are unlikely to affect outcomes such as progression to invasive disease and bladder cancer death. 

- The grouping between non-infiltrating tumors (Ta) and CIS is not correct: High grade Ta may have around a 10% risk of progression, while the CIS can reach 50% (figure 3). This analysis should be reconsidered only with Ta vs T1 if the number of patients with CIS is too small.

We have redone the analyses, now stratifying patients into Ta vs. T1 disease. This is reflected in the updated results and updated Figure 3 as well as in new Figure 4. The findings have remained unchanged. 

Reviewer #2: The authors have assessed the association of low- vs. guideline-recommended high-intensity cystoscopic surveillance with outcomes among patients with high-risk NMIBC analyzing a retrospective cohort study of Veterans Affairs patients (> 75) diagnosed with high-risk disease between 2005 and 2011 with follow-up through 2014

They found that patients with high-risk NMIBC undergoing low- vs. high-intensity cystoscopic surveillance underwent fewer transurethral resections, but did not experience an increased risk of progression or bladder cancer death

General comments

The evidence behind the frequency of cystoscopies in the follow-up of NMIBC is particularly poor and not supported by any level one evidence. Numerous studies including a previous one by the same group of authors (but mainly in low grade disease) have clearly outlined a major risk of overuse for various reasons.Therefore the attempt to correlate the frequency of cystoscopic examinations by the authors should be commended and addresses a real unmet need.

Although the results are nicely presented and the authors have done their very best to honestly analyze and criticize their findings nevertheless the reader is left with the impression that the low and high frequency groups may not have been similar, therefore weakening any robust conclusions.

1.The confounder and possible bias is patients with low intensity might have had a better prognosis and less recurrences (even the high risk NMIBC group is heterogenous as outlined by the authors in their discussion); so the treating urologist might have had the tendency to space follow-up.

Confounding and selection bias is a limitation of all retrospective cohort studies to some degree. We have attempted to limit this bias with propensity score adjustment. However, there are likely unmeasured confounders that influence an urologist’s surveillance cystoscopy practices. We have discussed many of these potential factors in the discussion section. We agree that additional research is needed to determine the optimal timing and frequency of surveillance cystoscopy among patients with high-risk NMIBC. 

2. After propensity score adjustment, patients who underwent low-intensity surveillance had almost 3-times fewer transurethral resections compared to those who underwent high-intensity surveillance: In high grade NMIBC, especially given the fact that these patients usually receive BCG leading to bladder inflammation, it is reasonable to assume that more frequent cystos led to a higher chance that the urologists would see something she/he didn't like, decided to err on side of caution and therefore resect, isn't it?

We agree that our findings are very plausible: looking more often likely lead to more abnormal cystoscopies and thus more transurethral sections. 

3.The surveillance window tended to be longer among patients who underwent low-intensity surveillance, because a lower proportion of them experienced a recurrence within the first 2 years (69 of 520 (13.3%) vs 655 of 1022 patients (64.1%); p<0.001). This is honestly difficult to reconcile and grasp. It seems that despite the matching, groups were imbalanced in terms of risk of recurrence/progression.

We agree that there is concern for residual imbalance between the groups and associated unmeasured confounding. We have discussed this extensively in the discussion section as well as in the response to item #4 below. 

4.Among patients with non-invasive disease (high grade Ta/CIS), low-intensity surveillance was associated with decreased risk of progression to invasive disease (T1 or T2) or bladder cancer death (CIn 19.3% vs. 31.4% at 5 years, p=0.002, Figure Panel 3B)- This is again very difficult to understand?? Why would any lower surveillance be associated with decreased risk of progression unless low intensity surveillance was performed in lower risk populations - the authors acknowledge that it is possible that urologists managed certain high-risk NMIBC patients with low-intensity surveillance based on intra-operative and patient characteristics not measured in their study. Urologists may have followed solitary or small high-risk lesions less intensely compared to multi-focal or larger tumors

We agree that this finding in particular is difficult to understand. This finding suggests that patients diagnosed with high-risk bladder cancer can undergo vastly different surveillance strategies and that those undergoing low-intensity surveillance may somehow represent a lower risk, “high risk” population. This further begs the need to determine optimal surveillance strategies for patients with high-risk disease, ideally in prospective randomized studies. There are likely tumor and patient characteristics that require more or less frequent surveillance cystoscopy than what is currently recommended by the AUA.

In conclusion, this is a nicely written paper on an important topic but the conclusions are not fully supported by the data presented and would benefit from being tempered.

Thank you. We have tempered our conclusion: 

“Patients with high-risk NMIBC undergoing low-intensity cystoscopic surveillance underwent fewer transurethral resections than those with high-intensity surveillance, but did not experience an increased risk of disease progression or bladder cancer death. These findings suggest that less intensive surveillance might be reasonable for patients with high-risk NMIBC. However, given the retrospective nature of this study, our findings are subject to unmeasured confounding. Thus, we do not advocate for lower intensity surveillance of high-risk NMIBC based on our findings. Rather, we believe our study provides a strong rationale for a future randomized trial to assess whether low-intensity surveillance of patients with high-risk NMIBC is comparable to high-intensity surveillance in terms of cancer control.”

---

## [Decision Letter · Decision Letter 1]

2 Mar 2020

The impact of low- versus high-intensity surveillance cystoscopy on surgical care and cancer outcomes in patients with high-risk non-muscle-invasive bladder cancer (NMIBC)

PONE-D-20-01946R1

Dear Dr. Rezaee,

We are pleased to inform you that your manuscript has been judged scientifically suitable for publication and will be formally accepted for publication once it complies with all outstanding technical requirements.

With kind regards,

Francisco X. Real

Academic Editor

PLOS ONE

Additional Editor Comments (optional):

Reviewers' comments:

Reviewer's Responses to Questions

**Comments to the Author**

1. If the authors have adequately addressed your comments raised in a previous round of review and you feel that this manuscript is now acceptable for publication, you may indicate that here to bypass the “Comments to the Author” section, enter your conflict of interest statement in the “Confidential to Editor” section, and submit your "Accept" recommendation.

Reviewer #1: All comments have been addressed

Reviewer #2: (No Response)

2. Is the manuscript technically sound, and do the data support the conclusions?

Reviewer #1: Yes

Reviewer #2: (No Response)

3. Has the statistical analysis been performed appropriately and rigorously? 

Reviewer #1: Yes

Reviewer #2: (No Response)

4. Have the authors made all data underlying the findings in their manuscript fully available?

Reviewer #1: Yes

Reviewer #2: (No Response)

5. Is the manuscript presented in an intelligible fashion and written in standard English?

Reviewer #1: Yes

Reviewer #2: (No Response)

6. Review Comments to the Author

Reviewer #1: After reading the new manuscript, all my comments addressed and adequately changed. I have no objection for publication.

Reviewer #2: (No Response)

7. PLOS authors have the option to publish the peer review history of their article (what does this mean?). If published, this will include your full peer review and any attached files.

Reviewer #1: No

Reviewer #2: No

---

## [Editor Report · Acceptance letter]

9 Mar 2020

PONE-D-20-01946R1 

The impact of low- versus high-intensity surveillance cystoscopy on surgical care and cancer outcomes in patients with high-risk non-muscle-invasive bladder cancer (NMIBC) 

Dear Dr. Rezaee:

I am pleased to inform you that your manuscript has been deemed suitable for publication in PLOS ONE. Congratulations! Your manuscript is now with our production department. 

With kind regards,

on behalf of

Dr. Francisco X. Real 

Academic Editor

PLOS ONE